# Out-of-Distribution Detection using Multiple Semantic Label Representations

**Gabi Shalev**
Bar-Ilan University, Israel
shalev.gabi@gmail.com

**Yossi Adi**
Bar-Ilan University, Israel
yossiadidrum@gmail.com

**Joseph Keshet**
Bar-Ilan University, Israel
jkeshet@cs.biu.ac.il

## Abstract

Deep Neural Networks are powerful models that attained remarkable results on a variety of tasks. These models are shown to be extremely efficient when training and test data are drawn from the same distribution. However, it is not clear how a network will act when it is fed with an out-of-distribution example. In this work, we consider the problem of out-of-distribution detection in neural networks. We propose to use multiple semantic dense representations instead of sparse representation as the target label. Specifically, we propose to use several word representations obtained from different corpora or architectures as target labels. We evaluated the proposed model on computer vision, and speech commands detection tasks and compared it to previous methods. Results suggest that our method compares favorably with previous work. Besides, we present the efficiency of our approach for detecting wrongly classified and adversarial examples.

## 1   Introduction

Deep Neural Networks (DNNs) have gained lots of success after enabling several breakthroughs in notably challenging problems such as image classification [12], speech recognition [1] and machine translation [4]. These models are known to generalize well on inputs that are drawn from the same distribution as of the examples that were used to train the model [43]. In real-world scenarios, the input instances to the model can be drawn from different distributions, and in these cases, DNNs tend to perform poorly. Nevertheless, it was observed that DNNs often produce high confidence predictions for unrecognizable inputs [33] or even for a random noise [13]. Moreover, recent works in the field of adversarial examples generation show that due to small input perturbations, DNNs tend to produce high probabilities while being greatly incorrect [11, 6, 17]. When considering AI Saftey, it is essential to train DNNs that are aware of the uncertainty in the predictions  [2]. Since DNNs are ubiquitous, present in nearly all segments of technology industry from self-driving cars to automated dialog agents, it becomes critical to design classifiers that can express uncertainty when predicting out-of-distribution inputs.

Recently, several studies proposed different approaches to handle this uncertainty [13, 25, 23, 19]. In [13] the authors proposed a baseline method to detect out-of-distribution examples based on the models' output probabilities. The work in [25] extended the baseline method by using temperature scaling of the softmax function and adding small controlled perturbations to inputs [14]. In [23] it was suggested to add another term to the loss so as to minimize the Kullback-Leibler (KL) divergence between the models' output for out-of-distribution samples and the uniform distribution.

Ensemble of classifiers with optional adversarial training was proposed in [19] for detecting out-of-distribution examples. Despite their high detection rate, ensemble methods require the optimization of several models and therefore are resource intensive. Additionally, each of the classifiers participated in the ensemble is trained independently and the representation is not shared among them.

In this work, we replace the traditional supervision during training by using several word embeddings as the model's supervision, where each of the embeddings was trained on a different corpus or with a different architecture. More specifically, our classifier is composed of several regression functions, each of which is trained to predict a word embedding of the target label. At inference time, we gain robustness in the prediction by making decision based on the output of the regression functions. Additionally, we use the L2-norm of the outputs as a score for detecting out-of-distribution instances.

We were inspired by several studies. In [26] the authors presented a novel technique for robust transfer learning, where they proposed to optimize multiple orthogonal predictors while using a shared representation. Although being orthogonal to each other, according to their results, the predictors were likely to produce identical softmax probabilities. Similarly, we train multiple predictors that share a common representation, but instead of using the same supervision and forcing orthogonality between them, we use different supervisions based on word representations. The idea of using word embeddings as a supervision was proposed in [8] for the task of *zero-shot learning*. As opposed to ours, their model was composed of a single predictor. Last, [39] found a link between the L2-norm of the input representation and the ability to discriminate in a target domain. We continue this thread here, where we explore the use of the L2-norm for detecting out-of-distribution samples.

The contributions of this paper are as follows:

- We propose using several different word embeddings as a supervision to gain diversity and redundancy in an ensemble model with a shared representation.
- We propose utilizing the semantic structure between word embeddings to produce semantic quality predictions.
- We propose using the L2-norm of the output vectors for detecting out-of-distribution inputs.
- We examined the use of the above approach for detecting adversarial examples and wrongly classified examples.

The outline of this paper is as follows. In Section 2, we formulate the notations in the paper. In Section 3 we describe our approach in detail. Section 4 summarizes our empirical results. In Sections 5 and Section 6 we explore the use of our method for detecting adversarial examples and wrongly classified examples. In Section 7 we list the related works, and we conclude the paper in Section 8.

## 2 Notations and Definitions

We denote by $\mathcal{X} \subseteq \mathbb{R}^p$ the set of instances, which are represented as $p$-dimensional feature vectors, and we denote by $\mathcal{Y} = \{1, \ldots, N\}$ the set of class labels. Each label can be referred as a *word*, and the set $\mathcal{Y}$ can be considered as a *dictionary*. We assume that each training example $(\mathbf{x}, y) \in \mathcal{X} \times \mathcal{Y}$ is drawn from a fixed but unknown distribution $\rho$. Our goal is to train a classifier that performs well on unseen examples that are drawn from the distribution $\rho$, and can also identify out-of-distribution examples, which are drawn from a different distribution, $\mu$.

Our model is based on word embedding representations. A *word embedding* is a mapping of a word or a label in the dictionary $\mathcal{Y}$ to a real vector space $\mathcal{Z} \subseteq \mathbb{R}^D$, so that words that are semantically closed have their corresponding vectors close in $\mathcal{Z}$. Formally, the word embedding is a function $\mathbf{e} : \mathcal{Y} \to \mathcal{Z}$ from the set of labels $\mathcal{Y}$ to an abstract vector space $\mathcal{Z}$. We assume that distances in the embedding space $\mathcal{Z}$ are measured using the *cosine distance* which is defined for two vectors $\mathbf{u}, \mathbf{v} \in \mathcal{Z}$ as follows:

$$d_{\cos}(\mathbf{u}, \mathbf{v}) = \frac{1}{2}\left(1 - \frac{\mathbf{u} \cdot \mathbf{v}}{\|\mathbf{u}\|\,\|\mathbf{v}\|}\right). \tag{1}$$

Two labels are considered semantically similar if and only if their corresponding embeddings are close in $\mathcal{Z}$, namely, when $d_{\cos}(\mathbf{e}(y_1), \mathbf{e}(y_2))$ is close to 0. When the cosine distance is close to 1, the corresponding labels are semantically far apart.

## 3 Model

Our goal is to build a robust classifier that can identify out-of-distribution inputs. In communication theory, robustness is gained by adding redundancy in different levels of the transmission encoding [20].

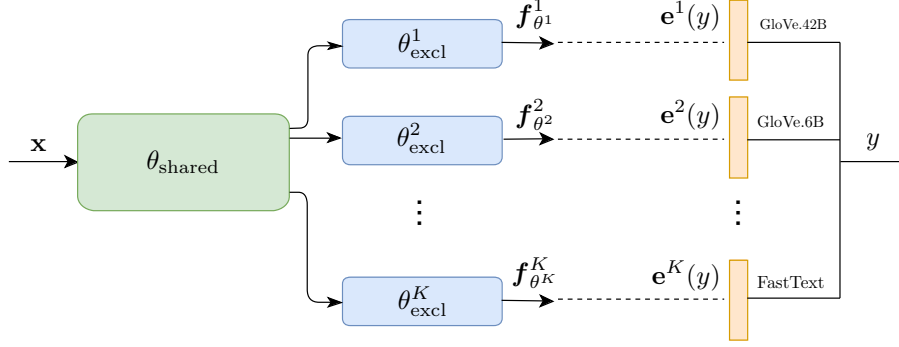

Figure 1: Our proposed $K$-embeddings model composed of $K$-predictors where each contains several fully-connected layers. The shared layers consisting a deep neural network.

Borrowing ideas from this theory, our classifier is designed to be trained on different supervisions for each class. Rather than using direct supervision, our classifier is composed of a set of regression functions, where each function is trained to predict a different word embedding (such as *GloVe*, *FastText*, etc.). The prediction of the model is based on the outcome of the regression functions.

Refer to the model depicted schematically in Figure 1. Formally, the model is composed of $K$ regression functions $\boldsymbol{f}^k_{\theta^k} : \mathcal{X} \to \mathcal{Z}$ for $1 \le k \le K$. The input for each function is an instance $\mathbf{x} \in \mathcal{X}$, and its output is a word embedding vector in $\mathcal{Z}$. Note that the word embedding spaces are not the same for the different regression functions, and specifically the notion of distance is unique for each function. Overall, given an instance $\mathbf{x}$, the output of the model is $K$ different word embedding vectors.

The set of parameters of a regression function $k$, $\theta^k = \{\theta_{\text{shared}}, \theta^k_{\text{excl.}}\}$, is composed of a set of parameters, $\theta_{\text{shared}}$, that is shared among all the $K$ functions, and a set of parameters, $\theta^k_{\text{excl.}}$, which is exclusive to the $k$-th function. Each regression function $\boldsymbol{f}^k_{\theta^k}$ is trained to predict a word embedding vector $\mathbf{e}^k(y)$ corresponds to the word which represents the target label $y \in \mathcal{Y}$. In the next subsections, we give a detailed account of how the model is trained and then present the inference procedure following the procedure to detect out-of-distribution instances.

## 3.1 Training

In classic supervised learning, the training set is composed of $M$ instance-label pairs. In our setting, each example from the training set, $\mathcal{S}_{\text{train}}$, is composed of an instance $\mathbf{x} \in \mathcal{X}$ and a set of $K$ different word embeddings $\{\mathbf{e}^1(y), \dots, \mathbf{e}^K(y)\}$ of a target label $y \in \mathcal{Y}$. Namely, $\mathcal{S}_{\text{train}} = \{(\mathbf{x}_i, \mathbf{e}^1(y_i), ..., \mathbf{e}^K(y_i))\}^M_{i=1}$.

Our goal in training is to minimize a loss function which measures the discrepancy between the predicted label and the desired label. Since our intermediate representation is based on word embeddings we cannot use the cross-entropy loss function. Moreover, we would like to keep the notion of similarity between the embedding vectors from the same space. Our surrogate loss function is the sum of $K$ cosine distances between the predicted embedding and the corresponding embedding vector of the target label, both from the same embedding space. Namely,

$$\bar{\ell}(\mathbf{x}, y; \theta) = \sum_{k=1}^{K} d_{\cos}(\mathbf{e}^k(y), \boldsymbol{f}^k_{\theta^k}(\mathbf{x})). \tag{2}$$

The cosine distance (or the cosine similarity) is a function often used in ranking tasks, where the goal is to give a high score to similar embedding vectors and a low score otherwise [9, 30].

## 3.2 Inference

At inference time, the regression functions predict $K$ vectors, each corresponds to a vector in a different word embedding space. A straightforward solution is to predict the label using *hard* decision over the $K$ output vectors by first predict the label of each output and then use a majority vote over the predicted labels.

Another, more successful procedure for decoding, is the *soft* decision, where we predict the label $y$ which has the minimal distance to *all* of the $K$ embedding vectors:

$$\hat{y} = \arg \min_{y \in \mathcal{Y}} \sum_{k=1}^{K} d_{\cos}(\mathbf{e}^k(y), \boldsymbol{f}_{\theta^k}^k(\mathbf{x})). \tag{3}$$

In order to distinguish between in- and out-of-distribution examples, we consider the norms of the predicted embedding vectors. When the sum of all the norms is below a detection threshold $\alpha$ we denote the example as an out-of-distribution example, namely,

$$\sum_{k=1}^{K} \|\boldsymbol{f}_{\theta^k}^k(\mathbf{x})\|_2^2 < \alpha. \tag{4}$$

This is inspired by the discussion of [39, Section 4] and motivated empirically in Section 4.3.

## 4  Experiments

In this section, we present our experimental results. First, we describe our experimental setup. Then, we evaluate our model using in-distribution examples, and lastly, we evaluate our model using out-of-distribution examples from different domains. We implemented the code using PyTorch [35]. It will be available under `www.github.com/MLSpeech/semantic_OOD`.

### 4.1  Experimental Setup

Recall that each regression function $\boldsymbol{f}_{\theta^k}^k$ is composed of a shared part and an exclusive part. In our setting, we used state-of-the-art known architectures as the shared part, and three fully-connected layers, with ReLU activation function between the first two, as the exclusive part.

We evaluated our approach on CIFAR-10, CIFAR-100 [18] and Google Speech Commands Dataset[1], abbreviated here as *GCommands*. For CIFAR-10 and CIFAR-100 we trained ResNet-18 and ResNet34 models [12], respectively, using stochastic gradient descent with momentum for 180 epochs. We used the standard normalization and data augmentation techniques. We used learning rate of 0.1, momentum value of 0.9 and weight decay of 0.0005. During training we divided the learning rate by 5 after 60, 120 and 160 epochs. For the *GCommands* dataset, we trained LeNet model [22] using Adam [16] for 20 epochs using batch size of 100 and a learning rate of 0.001. Similarly to [44] we extracted normalized spectrograms from the original waveforms where we zero-padded the spectrograms to equalize their sizes at $160 \times 101$.

For our supervision, we fetched five word representations for each label. The first two word representations were based on the *Skip-Gram* model [29] trained on Google News dataset and One Billion Words benchmark [5], respectively. The third and forth representations were based on GloVe [36], where the third one was trained using both Wikipedia corpus and Gigawords [34] dataset, the fourth was trained using Common Crawl dataset. The last word representations were obtained using Fast-Text [28] trained on Wikipedia corpus. We use the terms 1-embed, 3-embed, and 5-embed to specify the number of embeddings we use as supervision, i.e., the number of predicted embedding vectors. On 1-embed and 3-embed models we randomly pick 1 or 3 embeddings (respectively), out of the five embeddings.

We compared our results to a softmax classifier (*baseline*) [13], ensemble of softmax classifiers (*ensemble*) [19] and to [25] (*ODIN*). For the ensemble method, we followed a similar approach to [19, 24] where we randomly initialized each of the models. For ODIN, we followed the scheme proposed in by the authors where we did a grid search over the $\epsilon$ and $T$ values for each setting. In all of these models we optimized the cross-entropy loss function using the same architectures as in the proposed models.

### 4.2  In-Distribution

**Accuracy**    In this subsection, we evaluate the performance of our model and compare it to models based on softmax. Similar to [25], we considered CIFAR-10, CIFAR-100 and *GCommands* datasets

Table 1: Accuracy on in-distribution examples and semantical relevance of misclassifications.

| Dataset | Model | Accuracy | Avg. WUP | Avg. LCH | Avg. Path |
|---------|-------|----------|----------|----------|-----------|
| *GCommands* | Baseline | 90.3 | 0.2562 | 1.07 | 0.0937 |
| | 1-embed | 90.42 | **0.3279** | **1.23** | **0.1204** |
| | 3-embed | 91.04 | 0.3215 | 1.22 | 0.1184 |
| | 5-embed | **91.13** | 0.3095 | 1.19 | 0.1141 |
| | Ensemble | 90.9 | 0.2206 | 0.96 | 0.0748 |
| CIFAR-10 | Baseline | 95.28 | 0.7342 | 1.7 | 0.1594 |
| | 1-embed | 95.11 | **0.741** | **1.73** | **0.1633** |
| | 3-embed | 94.99 | 0.7352 | 1.71 | 0.1609 |
| | 5-embed | 95.04 | 0.7302 | 1.69 | 0.157 |
| | Ensemble | **95.87** | 0.733 | 1.71 | 0.1601 |
| CIFAR-100 | Baseline | 79.14 | 0.506 | 1.38 | 0.1263 |
| | 1-embed | 77.62 | 0.51 | 1.39 | 0.1277 |
| | 3-embed | 78.31 | 0.501 | 1.38 | 0.1251 |
| | 5-embed | 78.23 | **0.5129** | **1.4** | **0.1293** |
| | Ensemble | **81.38** | 0.5122 | **1.4** | 0.1291 |

as in-distribution examples. We report the accuracy for our models using $K = 1$, 3 or 5 word embeddings, and compare it to the baseline and to the ensemble of softmax classifier. Results are summarized in Table 1.

**Semantic Measure** Word embeddings usually capture the semantic hierarchy between the words [29], since the proposed models are trained with word embeddings as supervision, they can capture such semantics. To measure the semantic quality of the model, we compute three semantic measures based on WordNet hierarchy: (i) Node-counting on the shortest path that connects the senses in the is-a taxonomy; (ii) Wu-Palmer (WUP) [41], calculates the semantic relatedness by considering the depth of the two senses in the taxonomy; and (iii) Leacock-Chodorow (LCH) [21], calculates relatedness by finding the shortest path between two concepts and scaling that value by twice the maximum depth of the hierarchy. Results suggest that on average our model produces labels which have slightly better semantic quality.

### 4.3 Out-of-Distribution

**Out-of-Distribution Datasets** For out-of-distribution examples, we followed a similar setting as in [25, 13] and evaluated our models on several different datasets. All visual models were trained on CIFAR-10 and tested on SVHN[32], LSUN[42] (resized to 32x32x3) and CIFAR-100; and trained on CIFAR-100 and were tested on SVHN, LSUN (resized to 32x32x3) and CIFAR-10. For the speech models, we split the dataset into two disjoint subsets, the first contains 7 classes[2] (denote by SC-7) and the other one contains the remaining 23 classes (denote by SC-23). We trained our models using SC-23 and test them on SC-7.

**Evaluation** We followed the same metrics used by [13, 25]: (i) False Positive Rate (FPR) at 95% True Positive Rate (TPR): the probability that an out-of-distribution example is misclassified as in-distribution when the TPR is as high as 95%; (ii) Detection error: the misclassification probability when TPR is 95%, where we assumed that both out- and in-distribution have an equal prior of appearing; (iii) Area Under the Receiver Operating Characteristic curve (AUROC); and (iv) Area Under the Precision-Recall curve (AUPR) for in-distribution and out-of-distribution examples.

Figure 2 presents the distribution of the L2-norm of the proposed method using 5 word embeddings and the maximum probability of the baseline model for CIFAR-100 (in-distribution) and SVHN (out-of-distribution). Both models were trained on CIFAR-100 and evaluated on CIFAR-100 test set (in-distribution) and SVHN (out-of-distribution).

**Results** Table 2 summarizes the results for all the models. Results suggest that our method outperforms all the three methods in all but two settings: CIFAR-100 versus CIFAR-10 and CIFAR-

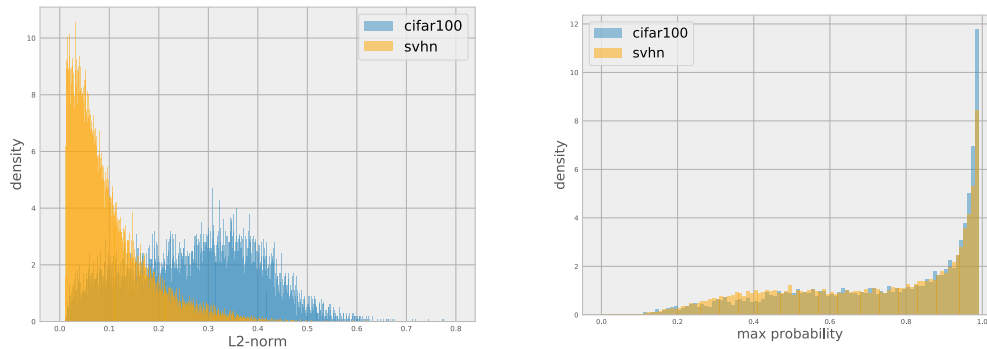

Figure 2: Distribution of the L2-norm of the proposed model with 5 word embeddings (left) and of the max probability of the baseline model (right). Both were evaluated on CIFAR-100 (in-distribution) and SVHN (out-of-distribution).

100 versus LSUN. This can be explained by taking a closer look into the structure of the datasets and the predictions made by our model. For these two settings, the in- and out-of-distribution datasets share common classes, and some of the classes of the in-distribution dataset appear in the out-of-distribution images. For example: the class *bedroom* in LSUN was detected as a *bed, couch,* and *wardrobe* in CIFAR-100 (56%, 18%, and 8% of the time, respectively); *bridge* of LSUN was detected as a *bridge, road*, and *sea* in CIFAR-100 (26%, 13%, and 9%); and *tower* of LSUN was detected as a *skyscraper, castle*, and *rocket* in CIFAR-100 (34%, 20%, and 6%), and there are many more. Similarly, CIFAR-10 and CIFAR-100 contain shared classes such as *truck*. Recall that the proportion of this class is 10% in CIFAR-10 and 1% in CIFAR-100. Hence, when the model is trained on CIFAR-100 and evaluated on CIFAR-10, it has 10% *in-distribution* examples a-priori. When the model is trained on CIFAR-10 and evaluated on CIFAR-100, it has 1% *in-distribution* examples a-priori.

## 5 Adversarial Examples

Next, we evaluated the performance of our approach for detecting adversarial examples [11]. Although it is not clear if adversarial examples can be considered as an out-of-distribution, we found that our method is very efficient in detecting these examples. Since our goal is detecting adversarial examples rather than suggesting a defense against them, we generated the adversarial examples in a black box settings.

We compared our method to an ensemble of softmax classifiers [19], both with $K = 5$ predictors, on the ImageNet dataset [7]. Both models are based on DenseNet-121 [15], wherein our model the $K$ regression functions are composed of three fully-connected layers with ReLU as described earlier. We omitted from ImageNet these classes which do not have "off-the-shelf" word representations and were left with 645 labels[3].

We generated adversarial examples in a black box setting using a third model, namely VGG-19 [38] with the *fast gradient sign method* and $\epsilon = 0.007$ [11], and measured the detection rate for each of the models. Notice that our goal was not to be robust to correctly classify adversarial examples but rather to detect them.

We used the method of detecting out-of-distribution examples to detect adversarial examples, results are in Table 3. We further explored the inter-predictor agreements on the predicted class across the $K$ predicted embeddings. The histogram of the differences between the maximum and the minimum rankings of the predicted label is presented in Figure 3. It can be observed that for our model the inter-predictor variability is much higher than that of the ensemble model. One explanation for this behavior is the transferability of adversarial examples across softmax classifiers, which can be reduced by using different supervisions.

Table 2: Results for in- and out-of-distribution detection for various settings. All values are in percentages. ↑ indicates larger value is better, and ↓ indicates lower value is better.

| In-Distribution | Out-of-Distribution | Model | FPR (95% TPR) ↓ | Detection Error ↓ | AUROC ↑ | AUPR-In ↑ | AUPR-Out ↑ |
|---|---|---|---|---|---|---|---|
| SC-23 (LeNet) | SC-7 | Baseline | 77.53 | 41.26 | 82.74 | 94.33 | 50.44 |
| | | ODIN | 71.02 | 38.01 | 85.49 | 95.11 | 57.7 |
| | | 1-embed | 70.64 | 37.8 | 85.85 | 95.21 | 58.08 |
| | | 3-embed | **67.5** | **36.24** | **87.34** | **95.91** | **59.97** |
| | | 5-embed | 69.23 | 37.09 | 86.93 | 95.74 | 58.97 |
| | | Ensemble | 72.73 | 38.85 | 85.99 | 95.69 | 50.71 |
| CIFAR-10 (ResNet18) | SVHN | Baseline | 7.19 | 6.09 | 97.2 | 96.35 | 98.05 |
| | | ODIN | 4.95 | 4.97 | 98.65 | 96.89 | 99.21 |
| | | 1-embed | **2.41** | **3.7** | **99.48** | **98.77** | **99.79** |
| | | 3-embed | 4.92 | 4.95 | 98.52 | 97.76 | 99.07 |
| | | 5-embed | 4.14 | 4.57 | 99.1 | 98.3 | 99.55 |
| | | Ensemble | 6.01 | 5.5 | 98.24 | 97.41 | 98.94 |
| CIFAR-10 (ResNet18) | LSUN | Baseline | 50.25 | 27.62 | 91.28 | 91.58 | 89.3 |
| | | ODIN | 41.8 | 23.39 | 90.35 | 96.38 | 75.07 |
| | | 1-embed | 26.11 | 15.55 | 95.37 | 95.81 | 94.85 |
| | | 3-embed | 29.2 | 17.09 | 95.07 | 95.63 | 94.38 |
| | | 5-embed | **22.98** | **13.99** | **96.05** | **96.72** | **94.86** |
| | | Ensemble | 46.16 | 25.58 | 92.93 | 94.1 | 77.57 |
| CIFAR-10 (ResNet18) | CIFAR-100 | Baseline | 58.75 | 31.87 | 87.76 | 86.73 | 85.83 |
| | | ODIN | 54.85 | 29.92 | 85.59 | 82.26 | 85.41 |
| | | 1-embed | 48.72 | 26.86 | 89.18 | 88.91 | 88.39 |
| | | 3-embed | 50.9 | 27.95 | 89.76 | 90.36 | 88.13 |
| | | 5-embed | **45.25** | **25.12** | **91.23** | **91.86** | **89.63** |
| | | Ensemble | 56.14 | 30.57 | 90.03 | 90.01 | 88.27 |
| CIFAR-100 (ResNet34) | SVHN | Baseline | 87.88 | 46.44 | 74.11 | 63.6 | 84.17 |
| | | ODIN | 76.64 | 40.82 | 79.86 | 68.22 | 90.1 |
| | | 1-embed | 75.79 | 40.39 | 81.82 | 71.52 | 90.1 |
| | | 3-embed | 74.74 | 39.87 | 82.57 | 75.01 | 90.4 |
| | | 5-embed | **60.14** | **32.57** | **87.42** | **77.95** | **93.56** |
| | | Ensemble | 85.92 | 45.46 | 79.1 | 69.23 | 89.3 |
| CIFAR-100 (ResNet34) | CIFAR-10 | Baseline | 77.21 | 41.1 | 79.18 | 80.71 | 75.22 |
| | | ODIN | 74.15 | 39.57 | 80.40 | 80.41 | 77.2 |
| | | 1-embed | 80.82 | 42.9 | 75.99 | 74.27 | 73.01 |
| | | 3-embed | 78.17 | 41.77 | 77.35 | 77.42 | 74.39 |
| | | 5-embed | 77.03 | 41.01 | 77.7 | 77.23 | 74.61 |
| | | Ensemble | **73.57** | **39.28** | **81.49** | **83.24** | **78.16** |
| CIFAR-100 (ResNet34) | LSUN | Baseline | 80.41 | 42.7 | 78.02 | 79.25 | 73.34 |
| | | ODIN | 79.88 | 42.44 | 78.94 | 80.22 | 73.31 |
| | | 1-embed | 80.99 | 42.99 | 76.41 | 75.08 | 74.02 |
| | | 3-embed | 81.08 | 43.03 | 74.88 | 72.21 | 72.38 |
| | | 5-embed | 80.87 | 42.93 | 76.08 | 75.3 | 72.67 |
| | | Ensemble | **79.53** | **42.26** | **79.05** | **92.61** | **74.06** |

We labeled an input as an adversarial example unless all the predictors were in full agreement. We calculated the detection rate of adversarial examples and the false rejection rate of legitimate examples. The ensemble achieved 43.88% detection rate and 11.69% false rejection rate, while the embedding model achieved 62.04% detection rate and 15.16% false rejection rate. Although the ensemble method achieves slightly better false rejection rate (3% improvement), the detection rate of our approach is significantly better (18% improvement). To better qualify that, we fixed the false rejection rate in both methods to be 3%. In this setting, the ensemble reaches 15.41% detection rate while our model reaches 28.64% detection rate (13% improvement).

## 6 Wrongly Classified Examples

Recall the embedding models were trained to minimize the cosine distance between the output vector and the word representation of the target label according to some embedding space. When we plot the average L2-norm of the models' output vectors as a function of the epochs, we have noticed

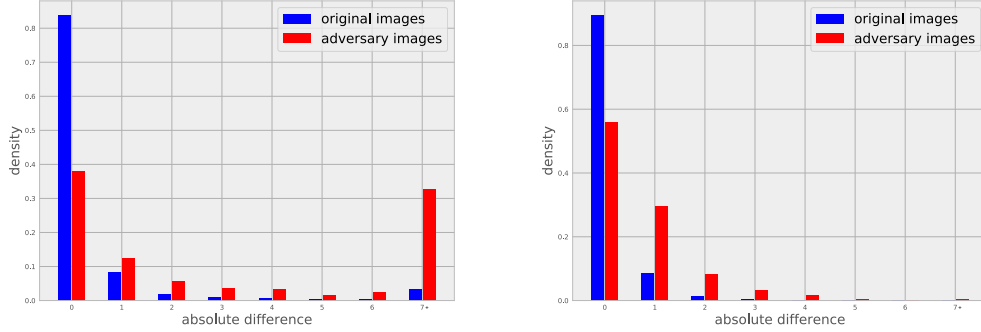

Figure 3: Histogram of the differences between the max and the min rankings of the predicted label in our model (left) and the ensemble model (right) for original and adversarial examples.

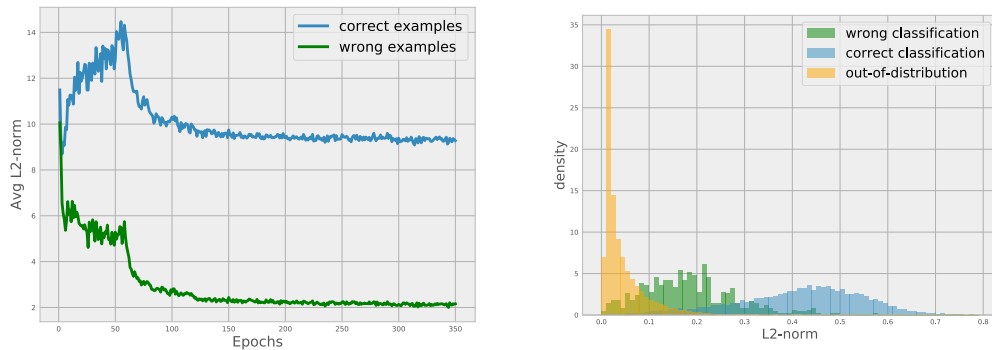

Figure 4: The average L2-norm for wrongly and correctly classified examples as a function of training epochs (left). Density of the L2-norm for wrongly, correctly classified, and out-of-distributions examples (right).

that the norm of the wrongly classified examples is *significantly smaller* than those of correctly classified examples. These finding goes in hand with the results in [39], which observed that lower representation norms are negatively associated with the softmax predictions, as shown in the left panel of Figure 4. Similarly, we observe a similar behavior for out-of-distribution examples as shown in the right panel of Figure 4. These findings suggest that we can adjust the threshold $\alpha$ accordingly. We leave this further exploration for future work.

## 7  Related Work

The problem of detecting out-of-distribution examples in low-dimensional space has been well-studied, however those methods found to be unreliable for high-dimensional space [40]. Recently, out-of-distribution detectors based on deep models have been proposed. Several studies require enlarging or modifying the neural networks [23, 37, 3], and various approaches suggest to use the output of a pre-trained model with some minor modifications [13, 25].

There has been a variety of works revolving around Bayesian inference approximations, which approximate the posterior distribution over the parameters of the neural network and use them to quantify predictive uncertainty [31, 27]. These Bayesian approximations often harder to implement and computationally slower to train compared to non-Bayesian approaches. The authors of [10] proposed to use Monte Carlo dropout to estimate uncertainty at test time as Bayesian approximation. More recently, [19] introduced a non-Bayesian method, using an ensemble of classifiers for predictive uncertainty estimation, which proved to be significantly better than previous methods.

Table 3: Results for in- and out-of-distribution detection, for ImageNet (in) and adversarial examples (out)

| Model | FPR (95% TPR) ↓ | Detection Error ↓ | AUROC ↑ | AUPR-In ↑ | AUPR-Out ↑ |
|---|---|---|---|---|---|
| Ensemble | 57.3 | 31.15 | 88.66 | **98.71** | 43.46 |
| 5-embed | **57.26** | **31.12** | **89.58** | 98.64 | **47.2** |

## 8 Discussion and Future Work

In this paper, we propose to use several semantic representations for each target label as supervision to the model in order to detect out-of-distribution examples, where the detection score is based on the L2-norm of the output representations.

For future work, we would like to further investigate the following: (i) we would like to explore better decision strategy for detecting out-of-distribution examples; (ii) we would like to rigorously analyze the notion of confidence based on the L2-norm beyond [39]; (iii) we would like to inspect the relation between detecting wrongly classified examples and adversarial examples to out-of-distribution examples.

## Footnotes

[1] `https://research.googleblog.com/2017/08/launching-speech-commands-dataset.html`

[2]Composed of the following classes: *bed, down, eight, five, four, wow*, and *zero*.

[3]In [8] the authors suggested to use random vectors as labels. We leave the exploration of this research direction for future work.

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
