[Reviews · NeurIPS 2018]

Reviewer 1



The problem of the study is out-of-distribution detection; we would like to detect samples that do not belong to the train distribution. To solve this problem, the authors propose reframing the prediction problem as a regression in which the sparse one-hot encoding of the labels in the prediction space is replaced by word embeddings. The presented method also has a flavour of ensemble methods in the sense that a base feature representation (Resnet for image prediction, Lenet for speech recognition) is used and then the feature branches off to multiple fully connected networks each of which generates an embedding vector as the prediction. Each fully connected network is responsible for generating vectors in a specific embedding space. A total of five embedding spaces are used (Skip-gram, GloVe, and FastText trained on independent text datasets). To detect out-of-distribution samples, they learn a lowerbound on the sum of the squared norms of the embedding predictions. If the sum falls below a threshold \alpha they classify it as an outlier. They provide a strong evaluation on the out-of-distribution problem using the literature measures and evaluations on several datasets comparing against the previous state-of-the-art (ODIN, ICLR 2018) and Deep Ensembles (NIPS, 2017). However, in the adversarial examples, they only provide a broad assessment and preliminary results within a limited testing environment. The authors also provide empirical results on the behaviour of wrongly classified examples within their pipeline, which indicates there may be a way to characterize uncertainty in the prediction in a reliable way (left for future work). Quality. The submission is technically sound, well-motivated and well-connected to the previous work. There are a few things that could be improved. i) For evaluation of ODIN, the authors fixed T=1000 and only find the optimal epsilon. Technically, they must grid search over all the temperatures (1, 2, 5, 10, 20, 50, 100, 200, 500, 1000) in the paper as well. ii) For evaluation of ODIN and DeepEnsemble, the authors compare against the network architectures that were used in the original papers, while the presented method is relying on Resnet-18 and Resnet-34. Both of the previous works do not strictly depend on a specific network architecture, therefore, their performance could be assessed using the same architectures as the presented method. While this difference in the base network seems insignificant, it is an important one for the evaluation since the topology of the network dictates different inductive biases and therefore different behaviour on the out-of-distribution samples. iii) For the adversarial examples, the authors extract adversarial examples from a VGG-19 network, while the base network architectures under study are DenseNet-121. Again, these two architectures are so drastically different in terms of the inductive biases that the performance of the networks under study will not provide any useful information for robustness against adversarial examples. However, the main selling point of the paper is around out-of-distribution detection and I'm not particularly worried about these experiments. I would be happy to increase my rating if the authors address (i) and (ii) properly. Clarity. The paper is well-written and easy to follow. However, the Related Work does not do justice to the literature. Novelty detection, open set recognition, prediction with abstention are the other keywords under which this problem is studied. Originality. As far as I'm familiar with the literature, the idea is novel and interesting. The connection to the previous work is well-justified and the work is a nice addition to the literature on this problem. Significance. The work is a clear improvement over the previous state-of-the-art (factoring the two issues that I mentioned earlier). I think the future work based on this idea could be a fruitful direction of research. ============================ Update after rebuttal. The authors adequately addressed my concerns. I have increased my overall score.

Reviewer 2



This paper is based on the idea of detecting an out-of-distribution example by training K regression networks to predict *distributed representations* of the target, and then declaring an outlier if the summed L2 norm of the K predicted embedding vectors is below a threshold (as in eq 4). In this paper the distributed representations are five different word embeddings of each target class, as per sec 4.1. The problem of out-of-distribution detection goes by many names, e.g novelty detection and "open set recognition". This is not sufficiently discussed in the paper. See e.g. https://www.wjscheirer.com/projects/openset-recognition/ and refs in [A] Finding the Unknown: Novelty Detection with Extreme Value Signatures of Deep Neural Activations. Schultheiss et al, German Conf of Pattern Recognition, 2017. The use of such distributed representations is reminiscent of Dietterich and Bakiri's work Solving Multiclass Learning problems via Error-Correcting Output Codes https://arxiv.org/pdf/cs/9501101.pdf (JAIR 1995), although the motivation is very different---there it is to make use of binary classifiers to tackle a multi-class problem. The idea is quite simple and the paper does not provide any theoretical basis, but from the experiments it seems that it outperforms the baseline, ensemble and ODIN competitor methods. One criticism is that it is not clear what exactly the 1-embed and 3-embed methods are -- is there optimization over which of the 5 embeddings you are using? If so this seems unfair. Please clarify the set up. I believe that ref [A] above is state-of-the-art for novelty detection. Thus I would like to know (in the rebuttal) the performance of this method on the authors' datasets. Strengths -- simple method, easy to implement. Appears to beat its competitors. Weaknesses -- lack of theory behind the method. Arbitrary nature of the distributed representations used -- for example would word embeddings be a good choice for predicting skin disease names from medical images? Why not other representations, or even say the higher layers in a deep network? Quality -- there is not much math that needs checking for technical correctness -- this paper is heavily empirical. Clarity -- clear enough. Originality -- as the authors acknowledge (line 124) their criterion in eq 4 is inspired by [39]. Significance -- out-of-distribution/novelty detection/open set recognition is an important problem. I am giving this paper a marginal accept for now, but the rebuttal will be very important wrt comparison to [A].

Reviewer 3



Summary: This paper suggests a new deep neural classifier that can effectively identify the out-of-distribution samples. Unlike most existing approaches which utilize the posterior distribution (usually, softamx function), the authors utilize the several word embeddings as the model's prediction with shared layers. In order to build a robust classifier, they propose a surrogate loss function which minimizes the cosine similarity between the predicted embedding and the corresponding embedding. Using the proposed inference and detection methods, they evaluate the proposed model on computer vision, and speech command detection tasks and compared it to previous methods. Strength: The paper is well written and the proposed idea is interesting. Weakness: However, I'm not very convinced with experimental results and I a bit doubt that this method would work in general and is useful in any sense. 1. The authors propose a new classification network, but I a bit doubt that its classification error is universally as good as the standard softmax network. It is a bit dangerous to build a new model for better detecting out-of-distribution samples, while losing its classification accuracy. Could the authors report the classification accuracy of the proposed classifier on ImageNet data? Some theoretical justifications, if possible, would be great for the issue. 2. The detection procedure (i.e., measuring the norm of the predicted embedding) is not intuitive and I am not convinced why it is expected to work. Could the authors provide more detailed explanations about it? 3. The baselines to compare are not enough, e.g., compare the proposed method with LID [1] which is one of the state-of-the-art detection methods for detecting adversarial samples. [1] Ma, X., Li, B., Wang, Y., Erfani, S.M., Wijewickrema, S., Houle, M.E., Schoenebeck, G., Song, D. and Bailey, J., Characterizing adversarial subspaces using local intrinsic dimensionality. In ICLR, 2018 4. Similar to Section 4.3, it is better to report AUROC and detection error when the authors evaluate their methods for detecting adversarial samples.